# Cathepsin S Upregulation Measured in the Peripheral Blood Mononuclear Cells Prior to Surgery Points to Postoperative Pain Development in Patients with Hip Osteoarthritis

**DOI:** 10.3390/diagnostics13101739

**Published:** 2023-05-15

**Authors:** Elena Tchetina, Kseniya Glemba, Anna Yunitskaya, Galina Markova, Maksim Makarov, Aleksandr Lila

**Affiliations:** 1Immunology and Molecular Biology Department, Nasonova Research Institute of Rheumatology, 34A Kashirskoe Shosse, 115522 Moscow, Russia; juni.a@yandex.ru (A.Y.); g.markova2010@yandex.ru (G.M.); 2Surgery Department, Nasonova Research Institute of Rheumatology, 34A Kashirskoe Shosse, 115522 Moscow, Russia; kseniyaglemba@yandex.ru (K.G.); ortopedniir@mail.ru (M.M.); 3Osteoarthritis Laboratory, Nasonova Research Institute of Rheumatology, 34A Kashirskoe Shosse, 115522 Moscow, Russia; amlila@mail.ru

**Keywords:** osteoarthritis of the hip joint, postoperative pain, gene expression, peripheral blood, cathepsin S

## Abstract

Disability caused by hip osteoarthritis has increased due to population aging, obesity, and lifestyle behaviors. Joint failure after conservative therapies results in total hip replacement, which is considered to be one of the most successful interventions. However, some patients experience long-term postoperative pain. Presently, there are no reliable clinical biomarkers for the prognosis of postoperative pain prior to surgery. Molecular biomarkers can be considered as intrinsic indicators of pathological processes and as links between clinical status and disease pathology, while recent innovative and sensitive approaches such as RT-PCR have extended the prognostic value of clinical traits. In light of this, we examined the importance of cathepsin S and proinflammatory cytokine gene expression in peripheral blood in addition to the clinical traits of patients with end-stage hip osteoarthritis (HOA) to predict postoperative pain development prior to surgery. This study included 31 patients with radiographic Kellgren and Lawrence grade III–IV HOA who underwent total hip arthroplasty (THA) and 26 healthy volunteers. Before surgery, a visual analog scale (VAS), DN4, PainDETECT, and the Western Ontario and McMaster Universities osteoarthritis index scores were used for pain and function assessment. Three and six months post-surgery, VAS pain scores of 30 mm and higher were reported. The intracellular protein levels of cathepsin S were measured using ELISA. The expression of the cathepsin S, tumor necrosis factor α, interleukin-1β, and cyclooxygenase-2 genes in peripheral blood mononuclear cells (PBMCs) was assessed using quantitative real-time RT-PCR. Pain persisted in 12 (38.7%) patients after THA. Patients who developed postoperative pain demonstrated significantly higher cathepsin S gene expression in the PBMCs and higher rates of neuropathic pain based on the DN4 testing compared to the other HOA subjects that were examined. No significant differences in proinflammatory cytokine gene expression were noted in either patient cohort prior to THA. The development of postoperative pain in patients with hip osteoarthritis might be associated with disturbances in pain perception, while increased expression of cathepsin S in the peripheral blood prior to surgery may serve as its prognostic biomarker and could be used in clinical settings to improve medical service for patients with end-stage hip OA.

## 1. Introduction

Osteoarthritis of the hip joint (HOA) is a major source of pain, disability, and socioeconomic expenditures worldwide [1]. Total hip arthroplasty (THA) is performed in cases of severe pain in end-stage HOA patients [2]. At present, THA is one of the most frequently performed surgeries, and this will continue until 2030 [3]. After surgery, many patients report significant reductions in pain, improvements in joint functioning and the range of motion, better alignment of the limb’s biological axis, reduced need for conservative therapy, and higher quality of life [4]. However, 7–23% of these patients continue to experience pain for a long time after surgery [5]. Age, gender, level of education, body mass index (BMI), comorbidities, depression, and radiographic parameters were considered as the reasons for unsatisfactory THA results [6,7]. At the same time, a patient’s pain is associated with their individual threshold for pain perception [8]. Therefore, it can be catastrophized due to a patient’s psychological traits, which should be considered in the prognosis of a THA outcome [9]. However, no reliable clinical markers predicting the results of THA associated with postoperative pain are presently known [10,11].

Recent advances in the understanding of the molecular mechanisms of pain development revealed new approaches for the identification of diagnostic biomarkers for pain prediction [12,13]. These mechanisms primarily involve neuroinflammation, which includes the activation of immune cells and the production of proinflammatory mediators and proteases [14]. The chronicity of inflammatory and neuropathic pain requires neuro–immune interaction [15]. In view of this, after a peripheral injury, microglial cells in the dorsal horn of the spinal cord proliferate and contribute to central sensitization and the generation of chronic pain by releasing proinflammatory cytokines [16]. On the other hand, cathepsin S, a cysteine protease that is a member of the papain superfamily, is located mainly in lysosomes of hematopoietic immune cells and is secreted into the extracellular environment [17]. As cathepsin S is stable at a neutral pH, it may contribute to extracellular proteolytic activity [18] and may be involved in the development of chronic pain by cleaving the membrane-bound pain-mediating chemokine fractalkine (CX3CL1) into a soluble form, which would further bind its receptor (CX3CR1) on microglial cells [19]. The latter activates the p38 MAPK pathway and releases proinflammatory mediators such as IL-1β [20]. In addition, cathepsin S can activate protease-sensitive receptor (PAR)-2, which stimulates Ca^2+^ release and induces inflammation [21].

Although studies related to the involvement of the above mediators as predictors of postoperative pain are limited, high protein concentrations of TNFα, IL6, and MMP-13 in the synovial fluid appear to be independent predictors of postoperative pain after knee arthroplasty [22]. However, as presurgical analysis of the synovial fluid is not easy, a search for biomarkers in the patient’s blood could be more promising [12]. A recent study demonstrated an association of cathepsins B and S activity at the protein level in serum with disease severity and joint inflammation in human knee OA [23]. However, only a moderate increase in cathepsin S activity in serum was observed in end-stage OA patients [23], who are commonly recommended joint replacement. At the same time, we demonstrated that high expression of cathepsin S, interleukin 1β (IL1β), and tumor necrosis factor α (TNFα) in peripheral blood mononuclear cells prior to surgery can serve as important biomarkers of POP development in patients with end-stage knee OA (KOA) [24].

Concurrently, it was not known whether these findings could be applied to hip OA, as it was suggested that the molecular pathophysiology of hip OA is not the same as that of knee OA [25]. This issue was primarily related to inflammatory processes that were demonstrated by the detection of proinflammatory molecules in patients’ serum and synovial tissues [26,27]. For example, several serum cytokines, such as IL8, fibroblast growth factor (FGF) 2, and monocyte chemotactic protein (MCP) 3, were differentially expressed in patients with HOA and KOA [26]. In addition, an assessment of the cytokine concentrations in the synovial membrane demonstrated higher amounts of IL4, IL10, and TNFα in HOA samples compared to KOA samples [27]. These dissimilarities suggest that prognostic biomarkers for POP after THA and total knee arthroplasty might be different. The identification of molecular predictors of surgical outcomes might reveal patients at risk of inadequate THA efficacy and promote the development of additional treatment strategies prior to surgery, aiming at an improvement in surgery results largely related to a reduction in postoperative pain (POP) development [28].

Therefore, the purpose of our study was to evaluate the significance of clinical indices and the expression of pain-related genes such as cathepsin S and proinflammatory cytokines in the peripheral blood of patients with HOA prior to surgery in search for prognostic biomarkers of POP development.

## 2. Materials and Methods

### 2.1. Ethics Statement

This study was performed in accordance with the Declaration of Helsinki. The study protocol (No. 32, dated 20 December 2018) was approved by the Local Human Research Ethics Committee of Nasonova Research Institute of Rheumatology for studies involving humans, and informed consent was obtained from all subjects.

### 2.2. Patients

The inclusion criteria for patients with end-stage HOA were as follows: A total of 31 unrelated subjects with primary OA of the hip joint who underwent primary total hip replacement surgery at the Nasonova Research Institute of Rheumatology between December 2018 and March 2020 were initially enrolled. No patients were excluded by the end of the study. The average age of the patients with OA was 61.3 ± 14.3 years (range: 46–79 years). These patients had radiographic Kellgren and Lawrence (K and L) HOA grades of III–IV, experienced constant pain during the last six months, and had limited mobility. The following NSAIDS were used as pain medications: meloxicam (15 mg/day) (*n* = 3), nimesulide (200 mg/day) (*n* = 9), diclofenac (200 mg/day) (*n* = 7), etoricoxib (60 mg/day) (*n* = 2), ketoprofen (100–200 mg/day) (*n* = 2), and celecoxib (200 mg/day) (*n* = 1). Patients were also treated with the symptomatic slow-acting drugs for osteoarthritis (SYSDOA) such as chondroitin sulphate (800 mg/day) (*n* = 1), glucosamine sulphate (1500 mg/day) (*n* = 6), and diacerein (100 mg/day) (*n* = 2). Medium-molecular-weight hyaluronate (2 mL, 3–5 doses/week) (*n* = 5) was also applied. All the patients with end-stage HOA fulfilled the criteria of the American College of Rheumatology regarding OA [29]. The exclusion criteria for OA patients were decompensated chronic diseases; active infectious diseases and compensated chronic infections; neurocirculatory disturbances of the lower extremities; opioid therapy prior to surgery; previous hip surgery; systemic inflammatory joint diseases; rheumatoid and secondary arthritis associated with reactive arthritis; gout; intra-articular fractures; ochronosis; acromegaly; Wilson disease; Padgett’s disease; primary synovial chondromatosis; hemochromatosis; chondrocalcinosis; aseptic necrosis of the femoral or tibial condyles; other abnormalities including renal diseases; thyroid, parathyroid, or other endocrinological diseases; diabetes mellitus; uncontrolled arterial hypertension; unstable angina; gastric or duodenal ulcers; vascular insufficiency; bleeding; thrombophlebitis; and the use of estrogen, progesterone, bisphosphonates, glucocorticoids, and alfacalcidol for women.

The inclusion criteria for control subjects were as follows: A total of 26 healthy individuals (average age: 62.8 ± 7.3 years, range: 42–74 years) were initially enrolled. No patients were excluded by the end of the study. The control subjects were of comparable age and gender to the examined patients with end-stage HOA and were recruited from the Moscow area. The exclusion criteria for control subjects included any degree of hip pain; rheumatoid arthritis; secondary arthritis associated with reactive arthritis; systemic inflammatory joint diseases; gout; pseudogout; Padgett’s disease; intra-articular fractures; ochronosis; acromegaly; hemochromatosis; Wilson disease; primary synovial chondromatosis; chondrocalcinosis; aseptic necrosis of the femoral or tibial condyles; any type of knee surgery; any abnormalities of bone metabolism, including diabetes mellitus; renal diseases; thyroid, parathyroid, or other endocrinological diseases; uncontrolled arterial hypertension; unstable angina; vascular insufficiency; gastric or duodenal ulcers; bleeding; and thrombophlebitis. Women who had taken drugs such as estrogen, progesterone, glucocorticoids, bisphosphonates, and alfacalcidol were not included in the study.

### 2.3. Demographic and Clinical Characteristics

The records of patient demographics, including age and gender, were obtained. The radiographic grade of HOA was determined via an analysis of anteroposterior radiographs of the pelvis taken with the patient in a supine position according to Kellgren and Lawrence and was based on the radiographic features of osteophytes on the joint margins, cystic areas, the sclerosis of subchondral bone, the narrowing of the joint space, and the altered shape of the femoral head [30]. The Western Ontario and McMaster Universities osteoarthritis index (WOMAC) was used to evaluate pain, stiffness, and physical function [31]. Nociceptive pain was evaluated using a visual analogue scale (VAS), and pain severity was evaluated using the Brief Pain Inventory (BPI) questionnaire [32]. The PainDETECT [33] and DN4 (Douleur Neuropathique en 4 Questions) [34] questionnaires were applied for neuropathic pain measurements, whereas anxiety and depression were revealed using the Hospital Anxiety and Depression Scale (HADS) [35]. The development of POP (≥30 mm on the VAS) was assessed during telephone surveys 3 and 6 months after THA.

### 2.4. Quantification of Cathepsin S Protein Levels in Peripheral Blood Mononuclear Cells of End-stage OA Patients and Healthy Controls

Peripheral blood (10 mL) was collected in vacutainers containing ethylenediaminetetraacetic acid (EDTA) (BDH, England). Peripheral blood mononuclear cells (PBMCs) were obtained using a Ficoll density gradient, washed twice in phosphate-buffered saline, and kept frozen at −80 °C until protein extraction. Cell lysates were acquired using a Cell Extraction Buffer (Invitrogen, Camarillo, CA, USA) supplemented with a Protease Inhibitor Cocktail (Sigma-Aldrich, Inc, St. Louis, MO, USA) and 1 mM PMSF (Sigma-Aldrich, Inc, St. Louis, MO, USA) according to the manufacturer’s instructions. The concentration of cathepsin S (ELH-CathepsinS) was determined in isolated PBMCs using a commercially available enzyme-linked immunosorbent assay (ELISA) kit (RayBiotech, Norcross, GA, USA) according to the manufacturer’s recommendations. The total DNA content in PBMC lysates was measured spectrophotometrically using a GeneQuant device (Amersham Biosciences, Watertown, MA, USA). The results were expressed per μg of DNA measured in the PBMC lysates.

### 2.5. Total RNA Isolation and Reverse Transcriptase (RT) Reaction

Total RNA was isolated from 100 μL of fresh whole blood using Extract RNA reagent (Evrogen, Moscow, Russia) in accordance with the manufacturer’s recommendations. The total RNA had an A260/290 > 1.9. The RT reaction was performed using an M-MLV RT kit containing Moloney Murine Leukemia Virus (M-MLV) Reverse Transcriptase, random hexanucleotide primers, and total RNA according to the manufacturer’s recommendations (Evrogen, Moscow, Russia).

### 2.6. Real-Time Quantitative PCR

The expression of the human genes cathepsin S (Hs00175407_m1), TNFα (Hs00174128_m1), IL-1β (Hs00174097_m1), and COX-2 (Hs00153133_m1) was assessed using TaqMan primers and probes (Applied Biosystems, Foster City, CA, USA). β-Actin served as an endogenous control. The quantification of gene expression was conducted using a QuantStudio 5 Real-Time PCR System (Applied Biosystems, Foster City, CA, USA) using a standard protocol [36]. A 1 μL volume of RT product was subjected to real-time PCR in a 15 μL total reaction mixture including 7.5 μL of TaqMan Universal PCR Master Mix (Applied Biosystems), 900 nM sense and antisense primers, a 50 nM probe, and a cDNA template. Relative mRNA expression was calculated using the ΔΔCT method, as described by the manufacturer (Applied Biosystems) [37]. Each PCR was performed in duplicate. Three “no template” controls were consistently negative for each reaction.

### 2.7. Statistical Analysis

Spearman’s rank correlations and the Mann–Whitney U-test were used to analyze non-normally distributed data, which were expressed as medians (Me) [quartiles, IQR]. A receiver operating characteristic (ROC) curve analysis included calculating the area under the curve (AUC) and the 95% confidence interval (CI). The diagnostic efficacy of the gene expression values was assessed using the sensitivity and specificity at the cut-off point. To compare percentages, a two-tailed Z-test for percentages was applied. Statistica for Windows and Statistical Package for the Social Sciences (SPSS) version 19 software (IBM, Armonk, NY, USA) were used for all the statistical analyses. *p*-values ≤ 0.05 were considered significant.

## 3. Results 

### 3.1. Clinical Characteristics of the Examined Patients with End-Stage HOA at Baseline

The average age of the examined patients with end-stage HOA (*n* = 31) was 61.3 ± 14.3 years (range: 46–79 years), whereas the average duration of the disease was 8.0 years (range: 2–50 years). These patients demonstrated K and L HOA grades of III to IV (grade III, 19 patients; grade IV, 12 patients) and increased body mass index (BMI) values (average: 28.8, range: 23.4–35.7). The total WOMAC scores fluctuated from 670 to 2450 (average: 1385), whereas the average total pain, total physical function, and total stiffness scores were 241.2 (range: 120–380), 873 (range: 440–1300), and 91.5 (range: 60–130), respectively.

All patients had suffered from permanent hip pain during the previous six months. The pain scoring according to the VAS demonstrated severe pain rates of 72.5 (range: 60–90) during movement and mild pain of 29 (range: 0–70) at rest in these patients with HOA. According to the Brief Pain Inventory (BPI) questionnaire, the average pain severity at that moment was 4.5 (range: 0–8) in the examined cohort. However, in the past 24 h, the worst average pain was self-assessed as a severe score of 7.0 (range: 0–9), while the lowest level of pain was reported as a mild score of 3.0 (range: 0–7). In addition, hip pain moderately interfered with walking and working abilities (average scores of 5.5 and 4.4, respectively). A neuropathic pain evaluation using the DN4 questionnaire revealed an average score of 1.16 (range: 0–2). The average score according to the PainDETECT questionnaire was 3.9 (range: 0–22), indicating one patient with neuropathic pain (score > 18). According to the Hospital Anxiety and Depression Scale (HADS), abnormal anxiety (score >11) was not observed, while 3 (10%) patients out of 31 exhibited borderline levels (scores of 8–10). In addition, 3 (10%) of 31 patients were depressed because they showed total scores of 11–13, whereas 14 (45%) of 31 patients had borderline values (scores of 8–10), according to the HADS depression scale.

Twelve patients with end-stage HOA developed POP 3 (mean VAS: 37.2 ± 9.0 mm) and 6 (mean VAS: 39.1 ± 8.3 mm) months post-surgery, while the others did not develop POP (*n* = 19). A comparative analysis of the baseline clinical parameters in both subgroups demonstrated significant differences in the values of several indices (Table 1). For example, patients who developed POP demonstrated higher BMI and DN4 values. In addition, the latter subgroup showed a trend towards older age (*p* = 0.056) and higher WOMAC (*p* = 0.06) and HADS anxiety (*p* = 0.07) scores. 

The average age of the healthy control subjects was 62.8 ± 7.3 years (range: 42–74 years). The control group consisted of 11 females and 15 males. They had normal erythrocyte sedimentation rates (2–30 mm/h), healthy BMI values (18.5–24.99 kg/m^2^), and no comorbidities. 

### 3.2. Baseline Expression of Genes in the PBMCs

Before surgery, the expression of the cathepsin S, IL1β, TNFα, and COX2 genes was significantly higher in all examined patients with end-stage HOA compared to the healthy controls. However, the patients who developed POP showed significantly higher expression of the cathepsin S gene (*p* = 0.02) compared to painless patients, whereas there were no intergroup differences in the expression of the IL1β, TNFα, and COX2 genes at baseline (Figure 1). The raw data on the baseline relative expression of the cathepsin S, TNFα, IL1β, and COX2 genes, as determined using real-time PCR in the blood of end-stage HOA patients who developed postoperative pain (*n* = 12) and painless subjects (*n* = 19) compared to healthy individuals, are presented in Appendix A.

### 3.3. Protein Concentration of Cathepsin S in PBMCs

The protein levels of cathepsin S were tested in the PBMCs of all patients with end-stage HOA. The twelve examined end-stage patients who developed postsurgical pain demonstrated significantly higher (*p* = 0.01) cathepsin S amounts compared to the pain-free subjects (Figure 2).

### 3.4. Correlations of Gene Expression with Demographic and Clinical Traits

A Spearman rank correlation analysis revealed a strong positive relationship between BMI and pain according to the VAS at rest and during movement as well as with the total WOMAC score; the PainDETECT scores correlated with BPI pain severity and WOMAC stiffness (Table 2). The anxiety and depression scores from the HADS questionnaire were also positively correlated. 

We also performed a Spearman rank correlation test on the gene expression values of cathepsin S and IL-1β and found significant (*p* < 0.05) positive correlations with the PainDETECT score (r = 0.483 and 0.751, respectively). IL1β gene expression was significantly positively correlated with the WOMAC pain score (r = 0.560). The expression of the cathepsin S gene was significantly (*p* < 0.05) positively correlated with the expression of the IL1β (r = 0.638) and COX2 (r = 0.433) genes. A Spearman rank correlation analysis of cathepsin S gene and protein expression also revealed a significant positive correlation (r = 0.637). In addition, we observed a tendency (*p* = 0.07) towards a positive correlation between IL-1β and COX2 gene expression (r = 0.323). Another non-significant negative correlation (*p* = 0.07) was noted between IL-1β gene expression and the HADS depression score (r = −0.325). 

To assess the prognostic value of cathepsin S gene expression, we performed an ROC curve analysis (Figure 3), which confirmed a statistically significant relationship between the expression of this gene before THA and the likelihood of developing POP. The threshold value of the expression of the cathepsin S gene was 5.69 (AUC = 0.741; sensitivity: 0.667; specificity: 0.632; 95% CI: 0.569–0.914; *p* = 0.02).

## 4. Discussion

The identification of predictors of pain, as a part of an international initiative promoted by multiple organizations including the WHO [38], might help identify patients at higher risk of moderate to severe postoperative pain and consequently improve pain management and patient satisfaction [39]. This issue is specifically important in the case of THA, where postoperative analgesia is responsible for up to 20% of the pharmaceutical costs [40]. Here, we found that the development of POP after THA can be predicted using clinical and molecular approaches, primarily the assessment of the gene expression of cathepsin S in the blood prior to surgery. The high prognostic value of this gene expression in the development of POP was confirmed by a large area under the ROC curve (AUC = 0.741). Our present results support previous observations [24] on the prognostic importance of cathepsin S gene expression for the prediction of POP development prior to surgery in patients with KOA and indicate the versatility of this biomarker in relation to surgery outcome expectations for end-stage OA patients.

In contrast, the gene expression of the proinflammatory cytokines TNFα, IL-1β, and COX2 demonstrated no significant differences between the examined patient cohorts. This observation demonstrates the differences in the prognostic validity of POP molecular biomarkers in patients with HOA and end-stage KOA subjects. The latter revealed significantly higher expression of IL1β and TNFα genes in individuals who developed POP compared to pain-free patients before surgery [24]. Our data are consistent with the results of other studies, which noted that the destruction of the hip and knee in OA is caused by different pathophysiological mechanisms based on an analysis of the contribution of different populations of immune cells and cytokine profiles in the synovial membranes of patients with KOA and HOA [26].

However, we did not observe any association between the level of preoperative pain and the development of pain after surgery, in contrast to previous studies on HOA that observed a link between the long-term use of opioids for pain relief before surgery and POP development [41]. On the other hand, our results are consistent with reports demonstrating a higher likelihood of POP development associated with more severe preoperative OA disease [42] since we also observed a preoperative trend towards higher WOMAC scores in patients who developed POP after THA. We also found significantly higher BMI values in end-stage patients with HOA who demonstrated POP after THA compared with patients who were satisfied with the results of the therapy, supporting a previous observation that an elevated BMI reduces the efficacy of anti-inflammatory treatments aiming to reduce POP [43].

In addition, we observed significantly higher levels of neuropathic pain, according to the DN4 questionnaire, and a trend towards increased anxiety, according to HADS testing, before surgery in patients with HOA who developed POP, supporting previous findings on positive correlations between anxiety, pain, and joint dysfunction [44]. In addition, recent studies have shown that chronic fatigue reported before surgery was also associated with pain and opioid use after THA [45], whereas central-sensitization-related symptoms prior to surgery in patients with OA were often associated with the development of POP [46]. Moreover, depressed patients who underwent psychological therapy showed less POP and better function compared to a control group six months after THA [47]. The importance of central mechanisms in the development and maintenance of POP was also confirmed by the absence of a correlation between preoperative structural changes in the joint and pain intensity in OA [48,49]. At the same time, it was also noted that THA usually results in a long-term decrease in pain sensitivity due to changes in central sensitization, which is associated with an increase in the pressure sensitivity threshold and a decrease in temporal summation, an increase in the levels of interferon-inducible cytokine IP10, and a decrease in the IL8 concentration in the cerebrospinal fluid and plasma [48].

Our study has several limitations. We present only very limited data on the healthy subjects that were used as study controls. The fact that we performed an ELISA assay using cell lysates rather than secreted cathepsin S protein can be considered to be another limitation of our study. However, an intracellular cathepsin S protein assessment was only required for the confirmation of our cathepsin S gene expression evaluation, which we found to be a prognostic biomarker of postoperative pain in end-stage patients with HOA prior to surgery. Finally, we used a visual analog scale (VAS) rather than a numerical rating scale (NRS), which some authors consider more reliable [50,51], although many studies have demonstrated strong similarities between both scales [52]. In our studies, we used a VAS score for the assessment of patients’ pain because this index is required for pain assessment in OA studies [29] and because we needed to match our results with those observed in our previous study related to postoperative pain in patients with end-stage knee OA, where this method was applied. 

## 5. Conclusions

In summary, our preliminary study demonstrated that high baseline gene expression of cathepsin S measured in the peripheral blood of patients with end-stage hip OA prior to arthroplasty may serve as an important biomarker for the development of postoperative pain. The importance of clinical assessments of the WOMAC score, BMI, and the psychological traits that were previously noted in association with postoperative pain after THA was also confirmed. Further studies involving large cohorts of patients are required to verify our findings on the importance of preoperative molecular and clinical assessments for the prognosis of postoperative pain development. These data will contribute to a significant improvement in the management of patients with end-stage hip OA who are recommended to undertake total hip arthroplasty.

## Figures and Tables

**Figure 1 diagnostics-13-01739-f001:**
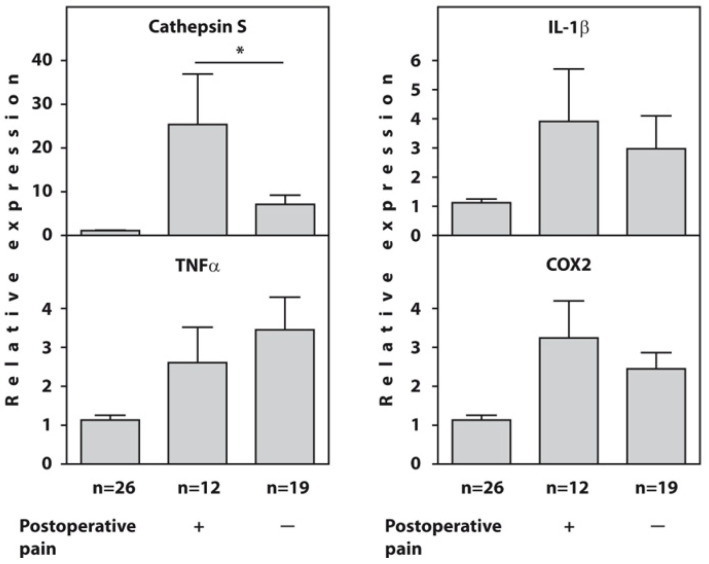
Baseline relative expression of the cathepsin S, TNFα, IL1β, and COX2 genes, as determined using real-time RT-PCR in the blood of end-stage HOA patients who developed postoperative pain (*n* = 12) and painless subjects (*n* = 19) compared to healthy individuals (*n* = 26). An asterisk (*) indicates significant differences (Mann–Whitney U-test) between subgroups of patients with OA.

**Figure 2 diagnostics-13-01739-f002:**
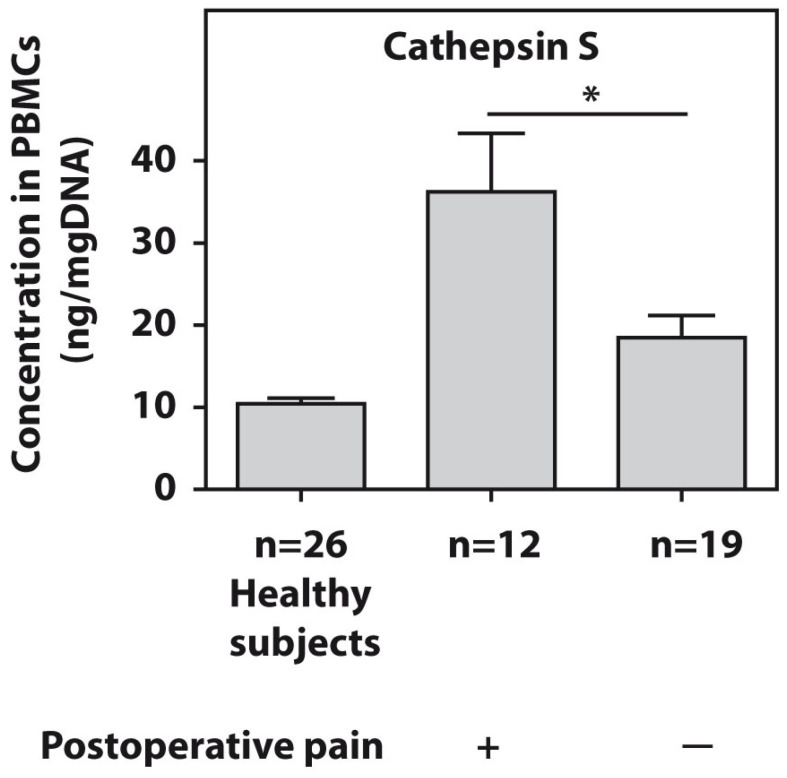
Baseline protein concentrations of cathepsin S, as measured using ELISA in PBMCs from patients with end-stage HOA who developed postoperative pain (*n* = 12) and pain-free subjects (*n* = 19). An asterisk (*) indicates significant differences (Mann–Whitney U-test) between the examined cohorts.

**Figure 3 diagnostics-13-01739-f003:**
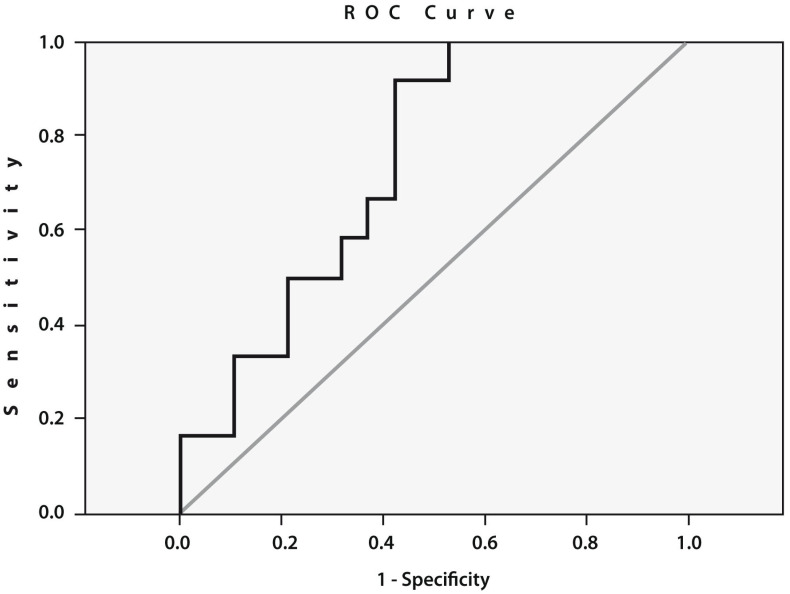
Receiver operating characteristic (ROC) curve for the expression of cathepsin S. Area under the curve (AUC) between the baseline cathepsin S gene expression in the peripheral blood of patients with end-stage HOA who developed postoperative pain (*n* = 12) and pain-free subjects (*n* = 19) (AUC = 0.741; sensitivity: 0.667; specificity: 0.632; 95% CI: 0.569–0.914; *p* = 0.02).

**Table 1 diagnostics-13-01739-t001:** Comparative characteristics of end-stage patients with HOA prior to THA.

	Patients Who Developed Pain 6 Months after Surgery (*n* = 12) Me [IQR]	Pain-Free Patients 6 Months after Surgery (*n* = 19) Me [IQR]	*p* (Mann–Whitney U-test)
Age, years	67 [63.5; 70.5]	49 [59; 67]	0.056
Gender, *n* (%)			
Females	9	5	
Males	3 (75)	14 (26)	0.076
Average Kellgren and Lawrence			
radiological stage, *n* (%)			
III	9 (82)	10 (53)	0.1
IV	3 (18)	9 (47)	0.1
Disease duration, years	4.5 [2.5; 7.5]	5 [4.5; 7]	0.3
Erythrocyte sedimentation rate (ESR), mm/h	9.5 [5.5; 15]	8 [6; 15]	1.0
Pain (VAS), mm	70 [70; 80]	70 [60; 80]	0.6
DN4 score	2 [1.5; 2]	1 [0; 1]	0.008
PainDETECT	5 [0.5; 6.5]	2 [1; 5]	0.6
HADS score, anxiety	5 [3; 6]	3 [1; 5]	0.07
HADS score, depression	8 [5.5; 9]	5 [4; 9]	0.14
BPI (pain severity)	3.6 [3.2; 5]	3.1 [3; 4]	0.13
WOMAC score	1295 [1175; 2045]	1170 [1050; 1350]	0.06
Comorbidities:			
BMI, kg/m^2^	29.7 [28.2; 33.0]	26.7 [25.2; 31.6]	0.04
Healthy, *n* (%): (BMI: 18.5–24.99 kg/m^2^)	1 (8)	3 (15)	0.5
Overweight, *n* (%): (BMI: 25–29.99 kg/m^2^)	5 (42)	10 (53)	0.5
Class 1 obesity, *n* (%): (BMI: 30–34.99 kg/m^2^)	4 (34)	5 (27)	0.6
Class 2 obesity, *n* (%): (BMI: 35–39.99 kg/m^2^)	2 (16)	1 (5)	0.3
Arterial hypertension (%)	33	31	0.9
Cardiovascular disease (%)	25	10.5	0.2

**Table 2 diagnostics-13-01739-t002:** Correlations of clinical parameters in patients with end-stage HOA before THA (*n* = 31).

	Pain at Rest	Pain during Movement	WOMAC Stiffness	TotalWOMAC	HADS Depression	BPI (Pain Severity)
BMI	0.94*p* = 0.01	0.76*p* = 0.04		0.58*p* = 0.01		
PainDETECT			0.87*p* = 0.01			0.92*p* = 0.002
HADS anxiety					0.83*p* = 0.01	
BPI (pain severity)			0.91*p* = 0.004

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
