# Peer review of "Cathepsin S Upregulation Measured in the Peripheral Blood Mononuclear Cells Prior to Surgery Points to Postoperative Pain Development in Patients with Hip Osteoarthritis"

_diagnostics, 2023, doi:10.3390/diagnostics13101739_

Round 1
Reviewer 1 Report
I was glad to review the work of the authors regarding this very interesting manuscript entitled "Cathepsin S upregulation measured in the peripheral blood 2 mononuclear cells prior to surgery points to postoperative pain 3 development in patients with hip osteoarthritis ". The manuscript is well-written and the incorporated tables and figures make the study easy to follow.
I strongly recommend acceptance for publication of the paper after minor changes.
1) In this study VAS scale was used. Why did you choose this scale and not the NRS scale?
2) "According to the literature, the NRS scale is usually chosen because compared to other pain intensity scales it is more preferable by patients, as well as in comparison to other pain scales (such as the Visual Analogue Scale, VAS), it is more sensitive in calculating the pain intensity changes that occur."
Add this information to the discussion section and consider citing the article:
https://pubmed.ncbi.nlm.nih.gov/33155461/
Author Response
RESPONSES TO REVIEWER 1
The authors are grateful to the reviewer for helpful and important comments. We have responded to all these comments as indicated below. Changes in the text are highlighted. We have made changed in accordance with the reviewer’s suggestions.
In this study VAS scale was used. Why did you choose this scale and not the NRS scale?
"According to the literature, the NRS scale is usually chosen because compared to other pain intensity scales it is more preferable by patients, as well as in comparison to other pain scales (such as the Visual Analogue Scale, VAS), it is more sensitive in calculating the pain intensity changes that occur." Add this information to the discussion section and consider citing the article: https://pubmed.ncbi.nlm.nih.gov/33155461/
We discussed the usage of VAS scale in our study in Discussion section and cited the recommended article [ref.#52].
Pain intensity can be measured using two scales: visual analog scale (VAS) and numerical rating scale (NRS). Although many studies have demonstrated strong similarities between both scales [53], some authors reported that NRS is more reliable scale [51,52]. In our studies we used VAS score for assessment of patients’ pain because this index is required for pain assessment in OA studies [29] and because we needed to match our results with that observed in our recent study related to postoperative pain in knee OA patients where the same method was applied. (Lines 371-377)
Reviewer 2 Report
I want to congratulate the authors to push the boundaries of medical science. Authors should be proud of their successful efforts to complete their research with honesty, creativity, humility, passion, and wisdom. As reviewers, we are here to improve how your novel data is displayed to the public. We are here to help you. Please, read and follow the recommendations.
Cathepsin S upregulation measured in the peripheral blood mononuclear cells prior to surgery points to postoperative pain development in patients with hip osteoarthritis.
Elena Tchetina
For MDPI DIAGNOSTICS
GENERAL: The whole text needs thoughtful revision especially to improve cohesion, precision, and comprehension. Some critical data is missing and some misinterpretations to correct. Author contribution: Which authors performed Blood extraction, RT-PCR, and ELISA?
ABSTRACT SECTION: The abstract need to be fully rewritten and reflect:
1. State of the art of the topic
2. Present a question or problem that needs to be resolved.
3. Then propose a potential solution, a new idea, and how (methodology) are you going to perform it.
4. Results (very brief)
5. Conclusions (remark on the importance of the new advancements)
INTRODUCTION SECTION: Readers must clearly understand what the authors have done in one go, they should not understand the facts by context. Precision is everything.
A very imprecise sentence as an example:
“We examined the importance of cathepsin S and pro-inflammatory cytokine gene expressions in peripheral blood as well as clinical traits of patients with end-stage hip osteoarthritis (HOA) before arthroplasty aiming identification of prognostic biomarkers for postoperative pain (POP) development.”
Another example:
However, 7-23% of these patients retain pain long time after surgery [5] while 35% have moderate or severe functional disturbances 5 years after THA [6]”.
First, these are the conclusions of two different studies (original & review), as such, the “35%” is completely unrelated to the “7-23%”. The statement should be separated into two sentences using “.” or “,” or “;”.
All statements must be referenced. If not, it is an assumption of the authors, and the wording would be different. As an example:
“At the same time, patient’s pain is associated with individual threshold for pain perception.”
I want to congratulate the authors on stating that they have previously done a similar study in KOA, and this current manuscript might enlighten in the hip OA.
Authors need to revise this particular section (lines 53-68), improve the transition from this section (clinical) to the next paragraph (molecular) and extend the state of the art of cathepsin S, the main topic of the study.
MATERIALS & METHODS SECTION: The inclusion & exclusion criteria are VERY promising but require a few changes.
HOA criterion: How many patients were initially enrolled?
· Inclusion: no comments
· Exclusion: How many of the initially enrolled subjects were excluded?
Control criterion: How many people were initially enrolled?
· Inclusion & exclusion criteria “serious illnesses” is too ambiguous. A description as precise and thoughtful as the HOA is required.
Quantification of cathepsin S Protein Levels: some doubts
It is known that Cathepsin S is synthesized and mainly secreted into the extracellular matrix. Low levels of Cathepsin S remain in lysosomes and phagosomes/ cytoplasmic vesicles. In line with these, ELISA was performed on cell lysates rather than cell secretion. This is a critical limitation of the study that should be mentioned in the Methodology or Results as well as in the Discussion section.
RESULTS SECTION
Section 3.1. There is no data on healthy patients. That is a severe limitation of the study.
It is interesting the comparison of Pain & Pain-Free patients. The authors should improve the transition into this paragraph and relate the choices and logic behind the comparison. On the other hand, the text does mention a separation between POP post -3 & 6 months, but the table is ambiguous about whether the data is combined or not. Please revise this section.
Section 3.2 These results are very interesting and should become the core of the manuscript. I encourage the authors to improve the section or even attempt to add experiments or other analyses. Nonetheless, there are some critical issues to address:
Graphs must display SD or SEM.
Differences between groups are narrow, to increase the likeness for the readers to use this data. I encourage the authors to add the RAW data in a supplementary file. The legend should reflect which method of * is being used (NEJM?)
There are about 4-6 patients in the HOA Pain-free group and 4 in the HOA with POP that present higher levels in all markers compared to the other patients. I might wonder if those are the same or different patients. Again, as I consider these data relevant, I encourage the authors to label each data point with a number /letter representing a specific patient to facilitate the comparison. Finally, I would suggest displaying the graphs in a 2x2 design.
Section 3.3 Control /Healthy Group is missing. Please display all data points. Is it SD or SEM? Y-axe is wrong, and X-Axe is missing information. Please reflect if this is cellular or secreted Cathepsin S.
Section 3.4 This is a critical section. The second paragraph (line 239) “In addition, we observed…” requires to be extended and precisely described. Words should not be carefully selected because this is not the Discussion nor Conclusion section”. Therefore, it is more like “We performed X correlation test between A & B and found a positive /negative correlation (r=?). Authors are encouraged to describe not only those correlations that succeeded but those that did not. I find it troubling not to find a single correlation with protein levels of cathepsin S.
Once changes are implemented there might be some correlations or differences that are no longer significant or the other way around. Please, literally use “tendency” or “non-significant” differences when required and “significative correlation or differences” when needed.
DISCUSSION SECTION: Please add a subsection on the limitations of the study.
DISCUSSION & CONCLUSIONS SECTION: Modify those statements linked to the comments above.
I feel qualified to assess the quality of an English-written text. I have experience as an English teacher for foreigners I acquired the Cambridge C2 Proficiency Certificate. Please follow my recommendations. As a tip, multiple free software currently corrects these non-orthographic errors.
Author Response
RESPONSES TO REVIEWER 2
The authors are grateful to the reviewer for helpful and important comments. We have responded to all these comments as indicated below. Changes in the text are highlighted. We have made changed in accordance with the reviewer’s suggestions.
The whole text needs thoughtful revision especially to improve cohesion, precision, and comprehension. Some critical data is missing and some misinterpretations to correct.
This has been corrected. We have responded to all comments. We significantly modified the text in accordance with the reviewer’s suggestions.
Author contribution: Which authors performed Blood extraction, RT-PCR, and ELISA?
This has been corrected: Blood extraction: A.K.Yu.; RT-PCR and ELISA: G.A.M; (Lines 391-392)
ABSTRACT SECTION: The abstract need to be fully rewritten and reflect: State of the art of the topic, Present a question or problem that needs to be resolved. Then propose a potential solution, a new idea, and how (methodology) are you going to perform it. Results (very brief). Conclusions (remark on the importance of the new advancements)
A very imprecise sentence as an example:
“We examined the importance of cathepsin S and pro-inflammatory cytokine gene expressions in peripheral blood as well as clinical traits of patients with end-stage hip osteoarthritis (HOA) before arthroplasty aiming identification of prognostic biomarkers for postoperative pain (POP) development.”
This has been corrected. We rewritten the Abstract in accordance with the Reviewer’s comments. (Lines 15-39)
INTRODUCTION SECTION: Readers must clearly understand what the authors have done in one go, they should not understand the facts by context. Precision is everything
This has been corrected. We revised Introduction section in accordance with the Reviewer’s comments.
However, 7-23% of these patients retain pain long time after surgery [5] while 35% have moderate or severe functional disturbances 5 years after THA [6]”.
First, these are the conclusions of two different studies (original & review), as such, the “35%” is completely unrelated to the “7-23%”. The statement should be separated into two sentences using “.” or “,” or “;”.
This has been corrected. We modified this part of the text in accordance with the Reviewer comment. (Lines 51-52)
All statements must be referenced. If not, it is an assumption of the authors, and the wording would be different. As an example:
“At the same time, patient’s pain is associated with individual threshold for pain perception.”
This has been corrected (Line 54). We provided references for all the statements.
Authors need to revise this particular section (lines 53-68), improve the transition from this section (clinical) to the next paragraph (molecular) and extend the state of the art of cathepsin S, the main topic of the study.
This has been corrected. We significantly revised Introduction (Section 1).
HOA criterion: How many patients were initially enrolled?
Exclusion: How many of the initially enrolled subjects were excluded?
Control criterion: How many people were initially enrolled?
This has been corrected. 31 patients with end-stage hip OA were initially enrolled. No patients were excluded by the end of the study. (Line 117). 26 healthy subjects were initially enrolled. No patients were excluded by the end of the study (Lines 140-141).
Inclusion & exclusion criteria “serious illnesses” is too ambiguous. A description as precise and thoughtful as the HOA is required.
This has been corrected. We added the exclusion criteria for control subjects included any degree of hip pain, rheumatoid arthritis; secondary arthritis associated with reactive arthritis, systemic inflammatory joint diseases, gout, pseudogout, Padgett’s disease, intraarticular fractures, ochronosis, acromegaly, hemochromatosis, Wilson disease, primary synovial chondromatosis, chondrocalcinosis, aseptic necrosis of femoral or tibia condilas, or any type of knee surgery; and any abnormalities of bone metabolism including diabetes mellitus; renal diseases; thyroid, parathyroid or other endocrinological diseases; uncontrolled arterial hypertension; instable angina; vascular insufficiency; gastric or duodenal ulcer; bleeding; or thrombophlebitis. Women who had taken drugs such as estrogen, progesterone, glucocorticoids, bisphosphonates, and alfacalcidol were not included in the study. (Lines 142-152)
Quantification of cathepsin S Protein Levels: some doubts
It is known that Cathepsin S is synthesized and mainly secreted into the extracellular matrix. Low levels of Cathepsin S remain in lysosomes and phagosomes/ cytoplasmic vesicles. In line with these, ELISA was performed on cell lysates rather than cell secretion. This is a critical limitation of the study that should be mentioned in the Methodology or Results as well as in the Discussion section.
This has been corrected. We added PBMCs to the title of section 2.4. However, the usage of PBMCs in our assessments was initially indicated in the titles of both Fig 2 and Section 3.3.
As we detected relative gene expression of cathepsin S in the PBMCs, to confirm our gene expression measurements, we determined protein concentrations of cathepsin S also in PBMCs (Lines169-170).
In addition, other studies demonstrated that serum levels of secreted cathepsin S cannot be considered as informative biomarker related to severity of the disease in the end-stage patients with OA [PMID: 25889265]. This issue has been noted in the Introduction section (Lines 79-83).
Section 3.1. There is no data on healthy patients. That is a severe limitation of the study.
This has been corrected. We added some data on healthy patients: The average age of the healthy control subjects was 62.8 ± 7.3 years (range 42–74 years). The control group consisted of 11 females and 15 males. They have normal erythrocyte sedimentation rate (2-30 mm/h), healthy Bone Mass Index (18.5-24.99 kg/m2), and no comorbidities. (Lines 250-253)
It is interesting the comparison of Pain & Pain-Free patients. The authors should improve the transition into this paragraph and relate the choices and logic behind the comparison. On the other hand, the text does mention a separation between POP post -3 & 6 months, but the table is ambiguous about whether the data is combined or not. Please revise this section.
This has been corrected. As it was indicated the data presented in the Table 1 was collected prior to surgery. Now we indicated this issue also in the text: Twelve patients with end-stage HOA developed POP after 3 (mean VAS 37.2±9.0 mm) and 6 (mean VAS 39.1±8.3 mm) months post-surgery while others did not develop POP (n=19). Comparative analysis of baseline clinical parameters in both subgroups demonstrated significant differences in values of several indices (Table 1). (Lines 241-244).
Section 3.2 These results are very interesting and should become the core of the manuscript. I encourage the authors to improve the section or even attempt to add experiments or other analyses. Nonetheless, there are some critical issues to address:
Graphs must display SD or SEM.
This has been corrected. Fig 1 and Fig 2 now display SEM.
Differences between groups are narrow, to increase the likeness for the readers to use this data. I encourage the authors to add the RAW data in a supplementary file.
This has been corrected. The raw data on baseline baseline relative expression of cathepsin S, TNFα, IL1β, and COX2 genes, determined by real-time PCR in the blood of end-stage HOA patients who either developed postoperative pain (n=12) or painless subjects (n=19), compared with healthy individuals is presented in Supplementary Table 1S. (Lines 262-264)
The legend should reflect which method of * is being used (NEJM?)
Figure legends for Fig. 1 and Fig.2 already contain indications that Asterisk (*) indicates significant differences (Mann-Whitney U-test) between subgroups of patients with OA. (Lines 273 and 285).
There are about 4-6 patients in the HOA Pain-free group and 4 in the HOA with POP that present higher levels in all markers compared to the other patients. I might wonder if those are the same or different patients. Again, as I consider these data relevant, I encourage the authors to label each data point with a number /letter representing a specific patient to facilitate the comparison.
This has been corrected. The raw data on baseline baseline relative expression of cathepsin S, TNFα, IL1β, and COX2 genes, determined by real-time PCR in the blood of end-stage HOA patients who either developed postoperative pain (n=12) or painless subjects (n=19), compared with healthy individuals is presented in Supplementary Table S. (Lines 262-264)
Finally, I would suggest displaying the graphs in a 2x2 design.
This has been corrected. Fig 1 is now displayed in a 2x2 design (Page 7).
Section 3.3 Control /Healthy Group is missing. Please display all data points. Is it SD or SEM? Y-axe is wrong, and X-Axe is missing information. Please reflect if this is cellular or secreted Cathepsin S.
This has been corrected. Fig.2 is arranged in accordance with the Reviewer’s recommendations. The data is presented as mean and SEM (Lines 281-282).
Section 3.4 This is a critical section. The second paragraph (line 239) “In addition, we observed…” requires to be extended and precisely described. Words should not be carefully selected because this is not the Discussion nor Conclusion section”. Therefore, it is more like “We performed X correlation test between A & B and found a positive /negative correlation (r=?). Authors are encouraged to describe not only those correlations that succeeded but those that did not. I find it troubling not to find a single correlation with protein levels of cathepsin S.
Once changes are implemented there might be some correlations or differences that are no longer significant or the other way around. Please, literally use “tendency” or “non-significant” differences when required and “significative correlation or differences” when needed.
This has been corrected. We have rewritten this part of the manuscript in accordance with the reviewer comments. (Lines 295-305). We added the data on significant positive correlation between cathepsin S gene and protein expressions (r=0.637) We also added data on two non-significant correlations.
DISCUSSION SECTION: Please add a subsection on the limitations of the study.
This has been corrected. The subsection on the limitations of the study was added (Lines 366-376)
I feel qualified to assess the quality of an English-written text. I have experience as an English teacher for foreigners I acquired the Cambridge C2 Proficiency Certificate. Please follow my recommendations. As a tip, multiple free software currently corrects these non-orthographic errors.
This has been corrected. We used Language Tool software (https://languagetool.org/ru) to correct all the errors in the text.
Round 2
Reviewer 2 Report
All requested changes have been implemented. No more comments.